# Toward an Estimation of the Optical Feedback Factor *C* on the Fly for Displacement Sensing

**DOI:** 10.3390/s21103528

**Published:** 2021-05-19

**Authors:** Olivier D. Bernal, Usman Zabit, Francis Jayat, Thierry Bosch

**Affiliations:** 1LAAS-CNRS, University of Toulouse, INP-ENSEEIHT, 31000 Toulouse, France; francis.jayat@toulouse-inp.fr (F.J.); thierry.bosch@toulouse-inp.fr (T.B.); 2Department of Electrical Engineering, National University of Sciences and Technology, NUST, Islamabad 44000, Pakistan; usman.zabit@seecs.edu.pk

**Keywords:** optical feedback interferometry, self-mixing, speckle, optical feedback factor, displacement measurement, non-uniform sampling

## Abstract

In this paper, a method based on the inherent event-based sampling capability of laser optical feedback interferometry (OFI) is proposed to assess the optical feedback factor *C* when the laser operates in the moderate and strong feedback regimes. Most of the phase unwrapping open-loop OFI algorithms rely on the estimation of *C* to retrieve the displacement with nanometric precision. Here, the proposed method operates in open-loop configuration and relies only on OFI’s fringe detection, thereby improving its robustness and ease of use. The proposed method is able to estimate *C* with a precision of <5%. The obtained performances are compared to three different approaches previously published and the impacts of phase noise and sampling frequency are reported. We also show that this method can assess *C* on the fly even when *C* is varying due to speckle. To the best of the authors’ knowledge, these are the first reported results of time-varying *C* estimation. In addition, through *C* estimation over time, it could pave the way not only to higher performance phase unwrapping algorithms but also to a better control of the optical feedback level via the use of an adaptive lens and thus to better displacement retrieval performances.

## 1. Introduction

Optical feedback interferometry (OFI), also referred to as the self-mixing (SM) effect in laser diodes (LDs) [1,2,3], has been widely investigated in recent decades as it results in a self-aligned and cost effective sensing system. The resolution of a stationary OFI-based displacement sensor depends on the employed signal processing techniques. Displacement measurement with a basic resolution of a half-wavelength ($\lambda_{0}$/2) can be easily achieved with an OFI sensor under a optical feedback regime by fringe counting [1]. The basic resolution can be improved by locking the laser phase to a half-wavelength [4] or by fringe duplication [5,6] or by utilizing phase unwrapping techniques. Different phase unwrapping techniques (based on time-domain OFI signal processing) have been proposed in the literature [7,8,9,10,11,12,13], providing accuracy from $\lambda_{0}$/8 to $\lambda_{0}$/60. Except for the fringe-locking method [4,14,15], for an accuracy exceeding $\lambda_{0}$/40, as well as for recent neural-network-based methods [16], these methods [8,9] require elaborate time-domain SM signal segmentations as well as estimations of key OFI parameters, such as optical feedback coupling parameter *C* [17]. In addition, the estimation of *C* can also be performed to allow an automatic OFI system setting via an adaptive optical lens [18]. As a result, assessing *C* over time appears to be necessary to achieve nanometric displacement reconstruction, especially in the presence of speckle.

Different methods have been devised to estimate *C* alone or conjointly with the linewidth enhancement factor α. In [18], a crude yet simple method based on the ratio of fringe sizes is used to ensure proper operation of the LD in the moderate feedback regime (1<C<4.6). In [19], a method to measure α together with *C* has been developed based on the Lang–Kobayashi theory [20]. From the shape of rising and decreasing SM fringes, two phase values, $\Phi_{13}$ and $\Phi_{24}$, corresponding to the phase difference between the fringe zero-crossing and discontinuity, can be extracted. Then, a unique set of $\left[\alpha ,C\right]$ suits $\left[\Phi_{13},\Phi_{24}\right]$. Even though this method can be used to characterize an LD, it might not be practical to be used during displacement measurement as it might suffer from non-linear displacements, high *C* values and speckle, which might lead to low frequency OFI signal modulation, thereby corrupting the zero-crossing detection. In [21], in order to alleviate this issue, it was demonstrated that from the frequency domain analysis of the phase of the LD with optical feedback $\Phi_{F}$, *C* can be estimated by calculating the ratio $\sum_{f\subset \Omega }\Phi_{F}/\sum_{f\subset \Omega }\sin\Phi_{F}$ with Ω denoting the spectral domain of validity, as defined in [21]. The range covered by this method is much larger than the previous one. However, it requires a proper estimation of $\Phi_{F}$, which requires both detecting all the fringes and unwrapping the SM signal. The unwrapping step [8] might be a difficult task to successfully complete due to noise. In [8], $\Phi_{F}$ unwrapping is also required as a first step. Then, in a manner similar to [7], a joint estimation of *C* and α is performed via an optimization procedure based on minimization of residual discontinuities in $\Phi_{F}$. As a result, this method is not suitable as such when the SM signal suffers from speckle.

Subsequently, here we propose a new open-loop approach that allows estimating *C* while retaining the inherent simplicity of OFI as the required hardware consists only of amplifying and acquiring the SM signal. Here, as presented in [22], we propose perceiving SM interferometers as inherent non-uniform sampling systems with their own embedded phase level-crossing detectors. Non-uniform sampling (NUS) approaches are often used in applications for which the retrieved information is sparse. Based on the NUS theory, we show that it is possible to retrieve *C*, which induces different quantization levels for the rising and decreasing fringes. Consequently, phase unwrapping techniques are no longer necessary. In addition, to recover *C* values either for sub-λ/2 displacements, for which a maximum of only one level crossing can be detected, or for SM signals disturbed by speckle, we propose adding a phase dither $\Phi_{d}$, either obtained by vibrating the sensor itself or by modulating the LD driving current, to the SM phase so that both the number of crossed levels as well as the rate of level crossings can be increased. This can allow a way to monitor *C* evolution to be devised and thus leads to the possibility of tuning a liquid or an adaptive lens to maintain the laser in the optimal *C* range for the chosen displacement reconstruction algorithm [18]. Note that dithering techniques have already been employed in OFI to achieve a high displacement resolution [23,24].

We will show that our method allows recovering *C* in the moderate feedback regime with a precision <3% (measured). In addition, we propose pushing this approach further to estimate *C* on the fly with a <5% precision when *C* is varying due to speckle (simulation). In the following section, Section 2, we present the non-uniform sampling theory applied to SM signals and how *C* can be estimated. We show how by applying a dithering signal, *C* can also be recovered in the presence of speckle. Then, in Section 3, the performances of the proposed method will be compared through extensive simulation to three other known methods [8,19,21]. This study will also encompass the effects of the sampling frequency of the data acquisition system, of the phase noise, of the amplitude and randomness of the displacement. In particular, it will show that knowing the *C* value over time is necessary to achieve nanometric displacement. In Section 4, different experimental test benches are then described and results are analyzed to assess the system performances. Both mechanical and electrical dithering methods are also used for comparison. Finally, conclusions are drawn in Section 5. It shows, in particular, that using dithering can allow retrieving varying *C* over time, which is necessary to achieve precision displacement better than 5 nm.

## 2. Proposed Method

### 2.1. OFI Overview

In OFI, a portion of the laser beam can be back-scattered from target placed at a distance $D_{0}$ from the laser (moving with displacement D(t)) and can thus re-enter the active laser cavity (Figure 1). This causes a mixing of generated and phase-shifted back-scattered beams. This “self-mixing” causes fluctuation in the optical output power (OOP) of the laser, denoted as P(t), given by [1]:(1)$$P\left(t\right)=P_{0}\left(1+m\cos\left(\Phi_{F}\left(t\right)\right)\right)$$
where $P_{0}$ is the emitted optical power under free-running conditions, *m* is the modulation index and $\Phi_{F}\left(t\right)$ is the laser output phase in the presence of feedback. $\Phi_{F}\left(t\right)$ is related to the laser output phase without feedback $\Phi_{0}\left(t\right)=4\pi D\left(t\right)/\lambda_{0}$ by [1,2]:(2)$$\Phi_{0}\left(t\right)=\Phi_{F}\left(t\right)+C\sin\left(\Phi_{F}\left(t\right)+\arctan\alpha \right)$$

Depending on *C*, the laser can operate in different regimes. SM sensing is generally performed under weak feedback regime (C<1), moderate feedback regime (1<C<4.6), or strong feedback regime ( C>4.6). However, a moderate feedback regime (1<C<4.6) is usually preferred as the apparently simple saw-tooth shaped SM fringes belonging to such a regime [25] intrinsically provide motion direction indication and require simplified SM fringe detection processing [26].

### 2.2. OFI as a Non-Uniform Sampling System

In the moderate feedback regime, based on (Equation 1) and (Equation 2), the information on $D\left(t\right)$ is completely enclosed within phase $\Phi_{F}$. It was shown in [22] that SM interferometers can be perceived as an inherent non-uniform sampling system with its own embedded phase level-crossing detector. A phase domain level crossing every 2π corresponding to the OOP discontinuities can thus be obtained (Figure 2). These discontinuities occur when $\Phi_{0}\left(t\right)=\Phi_{0}\left(k\right)$. However, it is important to note that, as shown in Figure 2, these phase levels $\Phi_{0}\left(k\right)$ are slightly different (by an amount denoted ΔΦ) for an increasing and decreasing $\Phi_{0}$ phase. These phase levels can thus be referred to as $\Phi_{0_{R}}$ and $\Phi_{0_{F}}$ when $\Phi_{0}$ is increasing or decreasing, respectively. They are completely defined by (Equation 2) with $\Phi_{F}$ as given in [27] whenever $\Phi_{F}$ has infinite slopes. ΔΦ can be expressed as a function of *C* [21]:(3)$$\Delta \Phi =2\left(\arccos\left(\frac{-1}{C}\right)+\sqrt{C^{2}-1}-\pi \right)$$

From (Equation 3) and Figure 3, it is clear that the function $\Delta \Phi \left(C\right)$ is bijective for C>0. As a result, estimating ΔΦ is the cornerstone of the proposed approach.

### 2.3. Proposed *C* Estimation Method

In most phase unwrapping algorithms, the first step consists of reconstructing a rough estimation $\Phi_{stair}$ of $\Phi_{0}$ by simply adding or subtracting 2π whenever a phase quantization level $\Phi_{0_{R}}$ and $\Phi_{0_{F}}$ is crossed, respectively. Based on the previous explanation, it is clear that $\Phi_{stair}$ does not take into account ΔΦ and hence contains some errors inherently. Here, we take advantage of this error to estimate ΔΦ.

To solve this issue, the non-uniform sampling approach allows one to look at it from a different perspective [22]. Instead of considering $\Phi_{0_{R}}$ and $\Phi_{0_{F}}$ to be different, they can be supposed to be equal if a virtual square-like displacement $D_{s}$ is added on top of *D*, with its rising (decreasing) edges corresponding to the change in direction. For the sake of clarity, this principle is illustrated by Figure 4. This figure shows that from the non-uniform sampling point of view, the sampled data (marked with red circle and black cross in Figure 4) are exactly the same. It thus ensures that the reconstructed signal from these samples is the same.

The peak-to-peak amplitude of this virtual square displacement should be equal to the equivalent displacement corresponding to ΔΦ. Thus, its amplitude $A_{s}$ can be expressed as:(4)$$A_{s}=\frac{1}{2}\frac{\lambda_{0}}{2}\frac{\Delta \Phi }{2\pi }=\frac{\lambda_{0}\Delta \Phi }{8\pi }$$

Consequently, ΔΦ can be indirectly estimated by measuring the amplitude of this virtual displacement $D_{s}$. More precisely, if ΔΦ is not taken into account, then harmonic distortion is bound to be generated in the reconstructed displacement due to the presence of this virtual square signal. By monitoring the amount of distortion, it is possible to estimate *C*. For instance, in the case of a sinusoidal displacement *D*, the third harmonic distortion amplitude, which should be equal to $4A_{s}/3\pi$, can be easily monitored. Figure 5 shows the performances that can be achieved based on this approach for different values of *C*. In accordance with Equation (Equation 2), it clearly shows that α has a negligible influence on the obtained results. In addition, *C* estimation accuracy expectedly depends on the displacement amplitude since the performance of the reconstruction algorithm based on NUS greatly improves with the number of crossed quantization levels [22].

However, while monitoring the third harmonic distortion amplitude is relatively convenient, it might not be effective in case of random displacements or in presence of speckle. In addition, the amplitude of the third harmonic not only depends on the *C* value but also on the interpolation algorithm that is used to reconstruct the displacement from the non-uniform samples. As a result, three different aspects of the proposed approach have been developed:$F_{1}$, estimation of *C* based on the direct amplitude of the 3rd harmonics and on (Equation 3).$F_{2}$, minimization of the amplitude of the 3rd harmonics by tuning the parameter ΔΦ/2 that is added (subtracted) to all the samples corresponding to the rising (decreasing) phase.$F_{3}$, minimization of the amplitude of all the harmonics, the frequency of which exceeds twice the frequency $f_{p}$ of the highest significant peak (>$\lambda_{0}/2$) by tuning the parameter ΔΦ/2 that is added (subtracted) to all the samples corresponding to the rising (decreasing) phase.

It is important to highlight once again that only $F_{3}$ can lead to relevant results in the case of non-periodic signals.

As a result, the proposed method to estimate *C* can be summarized by the block diagram shown in Figure 6 and the three following steps: (1) SM fringe detection, (2) displacement reconstruction based on non-uniform sampling approach [22] and (3) *C* estimation. During the first step, similarly to [22], from the SM signal, all the fringes should be detected to obtain all the time-phase pairs $\left[t_{n},\Phi_{n}\right]$, where $t_{n}$ corresponds to the time instant when the SM signal experiences a phase discontinuity. Since these pairs $\left[t_{n},\Phi_{n}\right]$ must be recorded, these $t_{n}$ values are quantized $Q\left(t_{n}\right)$ with a time resolution of $1/f_{s}$ (where $f_{s}$ is the sampling frequency of the data acquisition system) to generate non-uniform samples, $\left[Q\left(t_{n}\right),\Phi \left(t_{n}\right)\right]$. Then, during the second step, as shown in [28], the continuous time input signal can be reconstructed from these corrected samples if the quantization sampling rate of the input signal exceeds twice the input signal bandwidth. In addition, to be further processed, these $\left[Q\left(t_{n}\right),\Phi \left(t_{n}\right)\right]$ sets are usually fed to an interpolator to generate a uniformly sampled rate output signal. In this paper, the spline interpolator is used for simplicity. Finally, *C* can be estimated via the estimation of ΔΦ that can be performed directly either from the amplitude of the displacement’s third harmonic (in the case of a sinusoidal displacement) $F_{1}$ or from a minimization procedure of the harmonics by correcting the non-uniform time-phase pairs $\left[t_{n},\Phi_{n}\right]$ by adding or subtracting ΔΦ/2. Figure 7 shows the SM signal spectrum sampled at 10 MS/s and obtained for a 90 Hz 4 μm sinusoidal displacement and *C* = 3.3. In addition, Figure 8 emphasizes its corresponding reconstruction error with and without considering the estimated $C_{est}$ = 3.298 by the proposed approach $F_{2}$ and 10 MS/s sampling frequency. As expected, without considering *C*, the error is similar to a square signal with an amplitude of 120 nm (≈$\lambda_{0}\Delta \Phi \left(3.3\right)/8\pi =117$ nm) at the displacement signal frequency of 90 Hz, while by taking into account the estimated value of *C*, the root mean square (RMS) error is reduced down to approximately 1.9 nm. This clearly shows that once *C* has been estimated, it is then possible to efficiently reconstruct the displacement using the non-uniform sampling approach.

### 2.4. Analysis of the Impact of Variations of *C* on the Reconstructed Displacement

Up to now, *C* was considered to be constant. However, this is not usually the case as it depends on the amplitude of the displacement, on the remote target surface and on the laser beam spot. Consequently, the SM signal might suffer from the speckle phenomenon [29], which can cause signal amplitude fading and regime-change [30]. As a result, the *C* value may vary along the displacement. In this case, the phase gaps $\Delta \Phi_{R}$ and $\Delta \Phi_{F}$ between two successive $\Phi_{0,R}$ and $\Phi_{0,L}$ levels, respectively, are no longer constant and equal to 2π. Consequently, the displacement reconstruction cannot be accurate if these variations are not taken into account.

Before dealing with a randomly varying *C* value, we propose analyzing first the case where *C* is experiencing sinusoidal variations $C\left(t\right)=C_{0}+\delta C\left(t\right)$ with $\delta C\left(t\right)=C_{A}\sin\left(\omega_{c}t\right)$. This will give us more insight into the impact of these variations on the reconstructed displacement. In addition, in the case of periodic displacements, $\delta C\left(t\right)$ can be assumed to also be periodic with a period equal to a multiple of that of the displacement. As a result, such a variation can be described in terms of Fourier series.

Based on Equation (Equation 2), it can be shown that: (5)$$\begin{array}{ccc} {\left.\frac{\partial \Phi_{0}}{\partial C}\right|}_{\Phi_{F,R}} & = & \sqrt{1-\frac{1}{C_{0}^{2}}} \end{array}$$(6)$$\begin{array}{ccc} {\left.\frac{\partial \Phi_{0}}{\partial C}\right|}_{\Phi_{F,F}} & = & -\sqrt{1-\frac{1}{C_{0}^{2}}} \end{array}$$

Based on Equations (Equation 5) and (6), if $\delta C\left(t\right)$ are not taken into account, the reconstructed phase, assuming that $C=C_{0}$, ${\left.\Phi_{0}\right|}_{C=C_{0}}\left(t\right)$, is given by the following first order approximation:(7)$${\left.\Phi_{0}\right|}_{C=C_{0}}\left(t\right)=\Phi_{0}\left(t\right)-{\left(-1\right)}^{dir\left(t\right)}\delta C\left(t\right)\sqrt{1-\frac{1}{C_{0}^{2}}}$$
where dir depends on the target direction, which corresponds to dir = 0 and 1 for the case of Equations (Equation 5) and (6), respectively.

From Equation (Equation 7), it is clear that if $\delta C\left(t\right)$ are not taken into account by the displacement reconstruction algorithm, then error occurs as $\delta C\left(t\right)$ is interpreted as a phase change. It also shows that once $\delta C\left(t\right)$ is assessed, a first order approximation of $\Phi_{0}$ can be performed. Figure 9 illustrates this analysis with a sinusoidal displacement of 3 μm amplitude at $f_{0}$ = 25 Hz and $C=2.5+0.4\sin(2\pi \times 5f_{0}t)$. As expected using Equation (Equation 7), the spectrum (Figure 10) clearly shows a modification of the reconstructed displacement harmonics at 100 and 150 Hz.

It is therefore necessary to estimate *C* in order to correctly reconstruct the displacement. However, the proposed method (shown in Section 2.3) cannot be directly applied as the frequency at which the displacement direction changes is lower than the speed at which *C* changes. As a result, by directly applying the proposed method, only an average value of *C* can be obtained. Here, similarly to [22], in order to be able to apply the proposed approach, we propose adding a dithering signal that can be either generated by vibrating the LD itself or modulating the LD current at a frequency $f_{d}$ or using a phase modulator system based on the electro-optical properties of certain crystals (e.g., lithium niobate crystal) whose refractive indexes can be changed via the application of an electric field [31], to generate fringes and induce direction changes at a higher rate. Further, adding a dithering signal can be useful to retrieve sub-$\lambda_{0}$ displacement in a manner similar to approaches used in non-uniform sampling analog-to-digital converters [32,33]. Then, contrary to the previous case where *C* was supposed to be constant, in order to improve the time resolution regarding the estimation of *C*, it is necessary here to apply a spectral analysis using a sliding Hanning window along the SM signal. Together with the proposed method $F_{3}$, which relies on minimizing the harmonics amplitude, this will allow one to measure an average value of *C* across the Hanning window through the minimization of the power spectral density for frequencies higher than $f_{d}$.

Here, the proposed method extracts the information related to *C* by measuring the amount of distortion generated by the incorrect position of the phase quantization levels used for displacement reconstruction. This estimation is facilitated when the change of direction occurs as the error induced by ΔΦ is dominant. Consequently, to achieve a good estimation of *C*, it is necessary that the average rate of direction changes induced by the dithering signal should be at least twice the bandwidth of *C* variations to fulfill the Nyquist criterion [22]. However, the proposed method also relies on a time-frequency method based on applying spectral analysis over a sliding Hanning window. As a result, this implies a trade-off between the time and spectral resolutions. On the one hand, a wider sliding window will result in a better spectral resolution at the expense of time resolution and *C* estimation since the extracted value would correspond to the average value of *C* over the window. On the other hand, a shorter one will result in a better temporal resolution at the expense of the spectral resolution. Consequently, the method $F_{3}$ based on minimizing the harmonic content will greatly suffer from this reduced spectral resolution. Here, as a trade-off and as shown later, we have chosen to use a Hanning window that spans over $5/f_{d}$, which in return implies that the dithering frequency should be at least 10 times higher that the *C* bandwidth of interest in order to estimate *C* with a minimal error.

As previously mentioned, for large displacement and in case of non-cooperative remote target surfaces, *C* might also randomly vary due to the speckle phenomenon. In the particular case of periodic displacements, the *C* variation should also be periodic. This implies that *C* variations can be decomposed using Fourier series. As a result, the approach described in the previous section, Section 2.3, can be directly used. For randomly varying *C*, the proposed technique can still be applied if the parameters related to the Hanning window span and $f_{d}$ are in accordance with the required *C* bandwidth.

At last, in the case of a fast and sudden displacement (compared to the dithering frequency) of the target of few wavelengths, the *C* value could also rapidly change. As a result, if there is no phase direction change (induced by the dithering) during this event, the system would only be able to correctly estimate the *C* value before and after this sudden change and will only provide an interpolation in between, which might not be a correct estimation of *C* occurring during the fast change.

### 2.5. Phase Noise Effect

As the electrical noise has an indirect impact on the fringe detection as it makes it more difficult to detect them, different techniques have been proposed to alleviate it, such as filtering, Hilbert transform [10] and wavelet based algorithms [34]. In our case, the accuracy of the system is mainly dependent on the phase noise that directly affects the sets $\left[t_{n},\Phi_{n}\right]$, thereby altering the estimation of *C*. Phase noise will directly result in an increase of the noise floor. As a result, the optimization procedure can be stopped if the noise floor is reached during the process. In addition, the amplitudes of the harmonics induced by an incorrect estimation of *C* are also affected by the phase noise. More precisely, they will be underestimated since noise will tend to spread the harmonic power over a wider spectrum. Different noise sources can be identified [4]: the LD linewidth [35], the mechanical noise of the experimental set-up, the LD driving current noise and the temperature noise, which can both affect $\lambda_{0}$. Here, phase noise was simulated by adding noise to the target displacement.

Using spline interpolation to reconstruct the displacement from the NUS samples obtained via the SM signal, Figure 11 clearly shows that the floor noise increases with the phase noise δΦ but also that the amplitude of the harmonics generated by ΔΦ decreases with increasing δΦ. As a result, the proposed method accuracy will be directly limited by δΦ and its effects will be prominent for low *C* values as ΔΦ is smaller for low *C* values (Figure 3).

## 3. Simulation Results and Analysis

On top of the previously described simulation results, extensive simulations have been carried out to validate the proposed approach to estimate *C* and to compare its performances to other algorithms [8,19,21]. In order to ensure a fair comparison, the algorithms [8,21] employ here the same $\Phi_{F}$ reconstruction algorithm and all of them use the same fringe detection approach based on a basic derivative computation of the SM signal. Table 1 summarizes the results for SM signals obtained for a 3 μm sinusoidal displacements at 20 Hz sampled at $f_{s}$ = 1 MS/s with α = 3.6. These results are in accordance with [8,21]. The proposed approach based on minimization seems to offer performances similar to [8,19,21].

### 3.1. Influence of the Sampling Frequency and Displacement Amplitude

As the proposed approach relies on the NUS theory, it has its advantages and restrictions. In particular, both a higher $f_{s}$ and higher amplitude result in a more accurate estimation of *C* since the event can be dated more accurately and a higher number of quantization levels are involved in the process.

Figure 12, Figure 13 and Figure 14 show the influence of the sampling frequency $f_{s}$, of the displacement amplitude and of the SM phase noise on the achieved estimation of *C*, respectively. It clearly shows that $f_{s}$ should allow approximately 25 samples per fringe to be generated to date them with enough precision to achieve an approximation error of *C* lower than 2% error (for the 3rd flavour).

Figure 13 demonstrates the influence of the displacement amplitude on the *C* estimation error as expected.

### 3.2. Influence of Phase Noise

Here, the algorithms used to pre-process the SM signals (filtering, etc.) were modified to make them more resilient regarding noise. As a result, the obtained results without phase noise might be slightly different. Regarding phase noise, the *C* estimation error increases with phase noise with all the tested algorithms. Phase noise not only corrupts the fringe dating but also can lead to incorrect fringe detection (depending on the algorithm used for fringe detection). As a result, for the proposed algorithm, the phase noise will have a greater impact on the *C* estimation for low *C* value where ΔΦ is low (Figure Equation 2). Figure 14 also points out that the estimated *C* is always lower than the correct value, as expected for the proposed approach. Likewise, the estimated *C* value by the other approaches is also affected in the same manner.

### 3.3. Arbitrary Displacement

In the case of arbitrary displacement, the algorithm proposed in [21] cannot be applied as it expects the displacement power spectral density to be narrow-band. Similarly, [19] cannot be used as it expects sinusoidal displacement. For a similar reason, the approaches of $F_{1}$ and $F_{2}$ cannot be applied as there is no longer a 3rd harmonic that is properly defined. In addition, it might be necessary to use dithering in order to achieve a better estimation of *C*. In order to assess the performances, the input signal bandwidth is 100 Hz. Consequently, on the one hand, without dithering, $F_{3}$ is set to minimize the spectrum at frequencies greater than 100 Hz. On the other hand, with the dithering frequency set at 200 Hz with a 2 μm amplitude, $F_{3}$ is set to minimize the spectrum at frequencies greater than 100 Hz. Table 2 summarizes the results obtained for an arbitrary remote displacement (Figure 15) with and without dithering. It shows that dithering is required to correctly estimate *C*.

### 3.4. Speckle Affected SM Signals

Before processing arbitrarily varying *C* values, it is necessary to assess the performances of the proposed approach. We have highlighted in Section 2.4 that $f_{d}$ should be at least 10 times higher than $f_{C}$, where $f_{C}$ is the frequency of *C* variation. In accordance with this statement, Figure 16 clearly shows that to achieve an estimation of *C* with ΔC/C<5%, it is necessary that $f_{d}>10f_{C}$. To assess the influence of the Hanning window span on the *C* estimation, simulations of different window sizes show in Figure 17 that the optimum size is approximately $5/f_{d}$.

Figure 18 shows the case of a 10 Hz 10 μm displacement affected by arbitrary *C* variations with a limited bandwidth of 20 Hz. Even though the modulation index *m* from Equation (Equation 1) is proportional to *C*, it is assumed to be constant here for the sake of simplicity as it only affects the amplitude of the SM signal and thereby the fringe detection algorithm. The dither signal amplitude and frequency are set to be equivalent to 2 μm and $f_{d}$ = 200 Hz, respectively. The chosen Hanning window span is equal to 5/$f_{d}$. The *C* variations are correctly retrieved. The RMS error is approximately 2.4 nm with *C* taken into account to be compared to 10.6 nm otherwise. The peak error is approximately 5 nm and 33 nm for the former and latter, respectively. In addition, without dithering and *C* estimation, the RMS error is 44.9 nm similar to previous algorithm [7]. The reconstructed displacement spectrum shown in Figure 19 shows that a lower floor noise can be achieved and that artifacts around the dithering frequency can be removed through correct C(t) estimation.

## 4. Experimental Results

### 4.1. Experimental Setups

Two SM test benches (TB) (Figure 20) were developed to assess the performances of the proposed approach through two main test procedures. The aim of the first one (TB1 shown in Figure 20a) is to verify the ability of the proposed method to measure *C*. In this case, the target generates a sinusoidal displacement and an optical attenuator is used to modify *C*. The obtained values of *C* are compared to other methods and the reconstructed displacement error is computed. Table 3 summarizes the results. In the second test bench (TB2 shown in Figure 20b), the laser driving current was sinuoidally modulated to induce wavelength modulation. This will allow the generation of SM fringes to recover *C* not only in the case of sub-$\lambda_{0}$/2 displacement or even without any target displacement, but also in the case of speckle by generating a high-frequency dithering signal.

In TB1, the LD, driven by a constant injection current of 30 mA, is a Hitachi HL7851G emitting at $\lambda_{0}$ = 785 nm. The system benefits from the autofocus based on the liquid lens ARTIC 39N0 from Varioptic [18].

In TB2, the LD, driven by a average current of 25 mA, is a L1550P5DFB Telcordia emitting at $\lambda_{0}$ = 1550 nm. The driving current of this LD can be modulated up to ±5 mA to induce dithering either at 50 Hz or 1 kHz.

For both experiments, a piezoelectric transducer (PZT) from Physik Instrumente (P753.2CD) was used as a target positioned at 40 and 48 cm from the LD in TB1 and TB2, respectively. It was equipped with an internal capacitive feedback position sensor for direct-motion metrology with a 2 nm resolution. In order to generate speckle for large displacement, a loud speaker was used as the target while a shaker can be used to generate the dithering signal for TB1. The data were acquired by an NI USB 6251 data acquisition system operating at 1 Msamples/s with a 16 bit resolution. In the case of the TB2, when the LD was modulated at 1 kHz, the tektronix RTA4004 oscilloscope was used to acquired the data instead of the NI USB 6251. Prior to any measurements, the system phase noise for TB1 and TB2 was estimated to be approximately 0.136 rad and 0.192 rad, respectively, using the method described in [35].

### 4.2. *C* Constant without Dithering (TB1)

Using TB1, Table 3 summarizes the obtained performances with our proposed approach compared to [8,19,21]. For each of the six different *C* values, which cover the whole moderate feedback regime, twenty measurements were performed for a 1.5 μm sinusoidal displacement at 50 Hz. The repeatability of our method, which can be estimated via the average value of the obtained standard deviation σ, is similar or better than for the other algorithms. It is interesting to also note that the obtained values are lower than [8,21]. This can be explained by the fact that this approach is more affected by phase noise than the others, as shown in the previous sub-section (Figure 14). Nevertheless, Figure 21 shows that the measured SM signals (Table 3) and the simulated SM signal obtained with the estimated *C* value for a similar displacement amplitude are very similar. This shows a good correlation between the SM signal obtained using estimated *C* and the measured SM signal.

### 4.3. *C* Constant with Dithering (TB2)

In the TB2 configuration, the laser current was modulated either at 50 Hz or 1 kHz so as to obtain virtual displacement fringes (Figure 22). Based on the proposed method, the *C* value is extracted and compared to the one obtained with the same optical conditions but without laser current modulation and target vibration (similar to TB1). Contrary to TB1, the optical attenuator was removed here in order to avoid any parasitic reflections from it. These parasitic reflections combined with the laser driving current modulation could actually generate parasitic SM signals that might corrupt the extraction of the *C* value related to the target. As a result, to obtain different *C* values, the target was slightly shifted perpendicularly to the optical axis. For each setting, the SM signals were acquired 10 times except for the 1 kHz modulation configuration, for which only five acquisitions were performed due to a lack of automation. The results are summarized in Table 4. It shows that the results obtained with the proposed method based on dithering generated by current modulation are in good accordance with the proposed method without current dithering and also with other methods [18,19] where the SM fringes are generated by the vibrating target.

### 4.4. *C* Varying

As mentioned previously, the SM signal can be affected by speckle, which induces *C* variations. In the case where the *C* variation spectrum is lower than the target displacement spectrum, it might be possible to recover these variations by using the method presented in Sub-Section III.D. Figure 23 shows an SM signal affected by speckle for a target vibrating at 50 Hz suffering from low frequency small lateral displacements. It demonstrates that the proposed method can detect and assess the *C* variations. In addition, it also confirms that choosing a window size of five times the period of the displacement main frequency enables a good trade-off between time and amplitude estimation.

In the case for which the speckle frequency spectrum is comparable to that of the target displacement, the previous approach is not able to provide a correct estimation of *C* as it will result in an average estimation of *C*. Consequently, it is necessary to add a dithering signal to recover all the information. Figure 24 shows the measured results extracted from the SM signal acquired during 0.4 s in order to retrieve *C* while the SM signal is affected by speckle. In this case, the target is a loudspeaker vibrating at $f_{v}$ = 40 Hz with an amplitude of 2.4 μm, approximately, while the dithering vibration induced by a shaker is set at $f_{d}$ = 290 Hz with an amplitude of 1 μm, approximately. It clearly shows that *C* varied from 1.3 to 3.2 with the maximum and minimum appearing with a frequency of 40 Hz and *C* appearing to follow a 10 Hz pattern. As expected, this pattern appears to be periodic. It might be the result of the speckle induced by the loudspeaker vibration added to the one induced by the laser vibration.

In order to remove the impact of the mechanical dithering on *C* estimation in this case, it can be possible to generate it by modulating the laser driver current, as previously shown (Figure 20b). Here, the laser driver current was modulated at 5 kHz with an amplitude of ±3.6 mA. A shaker was used to vibrate at 170 Hz a piece of sandpaper to generate speckle within one period. Figure 25 shows the estimated *C*. Even though the trend appears to be correct, the algorithm used here to detect the fringes that is based on a basic derivative computation of the SM signal, is not able to correctly detect all the fringes occurring more particularly for low *C* values as the displacement reconstruction graph (Figure 25c) suggests it.

## 5. Conclusions

By interpreting OFI as a non-uniform event-based sampling system, as in [22], it was shown that in the moderate optical feedback regime, the phase quantization levels depend on *C* and that this property can be used to directly assess the value of *C*. Three different aspects of the method were presented. While the first two ($F_{1}$ et $F_{2}$) can be directly derived from the theoretical analysis of ΔΦ, they are both much less robust and versatile than $F_{3}$. Once *C* is correctly estimated, a better reconstruction of the displacement can be obtained even with the spline interpolator [22].

In addition, it was shown that the proposed method to assess *C* achieves similar simulated performances compared to [8,19,21]. The measured performances show that a <5% precision can be achieved.

Further, compared to other works, which assume either a sinusoidal displacement or an OFI signal devoid of speckle by adding a dithering signal, the proposed method based on $F_{3}$ can estimate *C* for arbitrary displacements and speckle affected OFI signals. It thus paves the way toward estimating *C* on the fly. It was shown in simulation that being able to estimate *C* in these conditions allows high-precision displacement reconstruction $\epsilon_{rms}<5$ nm, while based on both a relatively simple set-up and processing method even in the presence of speckle. Once *C* is estimated, during the first step of the unwrapping algorithms [7,8], the rough phase estimation should not only be obtained by simply adding or subtracting 2π when a discontinuity is detected for a moderate OFI regime, but also by taking into account any ΔΦ induced by *C* variations.

Up to now, the system’s main limitation is the bandwidth directly related to the dithering frequency either due to the use of a bulky system or due to the laser wavelength response to current modulation. This can be greatly improved by designing a very light weight laser head with an embedded PZT. Likewise, using electro-optical modulation appears to be also another promising way to achieve much higher bandwidths.

## Figures and Tables

**Figure 1 sensors-21-03528-f001:**
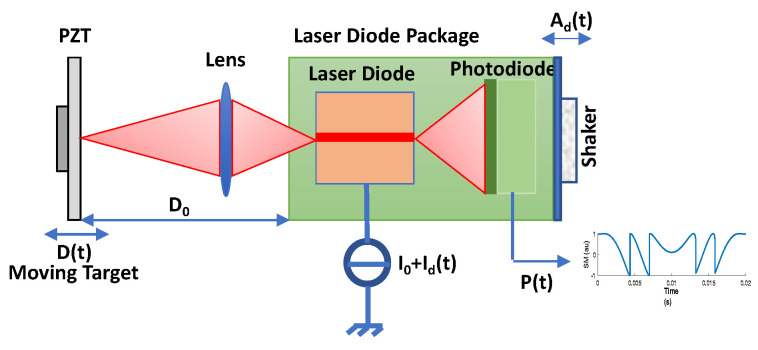
Self-Mixing displacement sensor set-up with a piezoelectric transducer (PZT) used as a target. A dithering signal can be added either via the laser drive current $I_{0}+I_{d}\left(t\right)$ or via a shaker vibrating the sensor itself with an amplitude $A_{d}\left(t\right)$.

**Figure 2 sensors-21-03528-f002:**
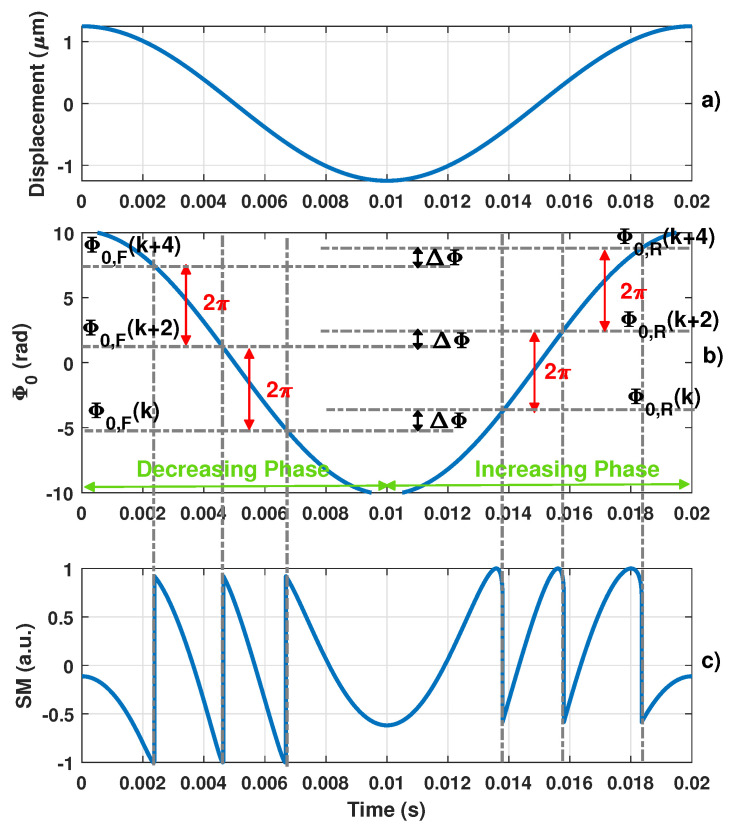
Simulated typical self-mixing signal showing hysteresis (**c**) obtained for (**a**) a 1.25 μm sinusoidal displacement, a laser wavelength $\lambda_{0}$ = 1550 nm and an optical coupling factor *C* = 2 with (**b**) its corresponding phase $\Phi_{0}$ and its phase quantization levels $\Phi_{0,F}$ and $\Phi_{0,R}$ for the decreasing and increasing phases, respectively.

**Figure 3 sensors-21-03528-f003:**
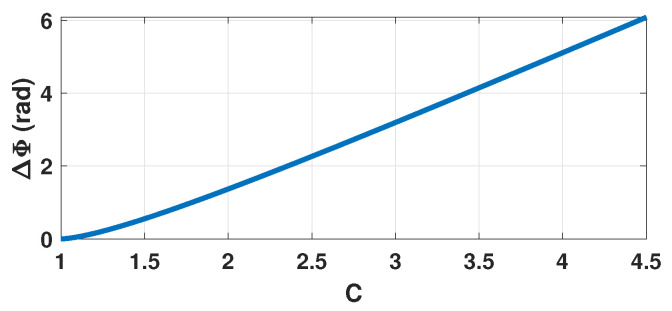
Simulated ΔΦ vs. *C* where ΔΦ denotes the phase quantization difference between the rising and decreasing fringes.

**Figure 4 sensors-21-03528-f004:**
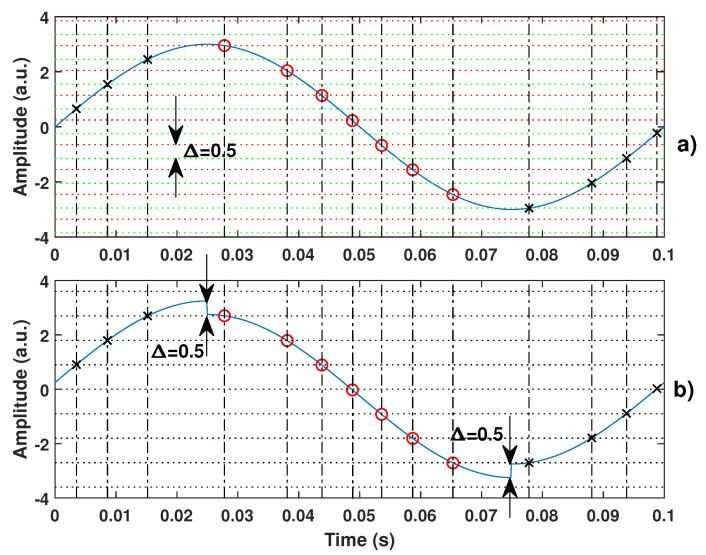
Illustration of the proposed equivalence between (**a**) the quantization levels which are different for rising (dash green) and decreasing (dash red) signals (by an amount here of 0.5 a.u.) and (**b**) where the quantization levels for rising and decreasing signals are the same due to addition of a square signal of 0.25 a.u. amplitude.

**Figure 5 sensors-21-03528-f005:**
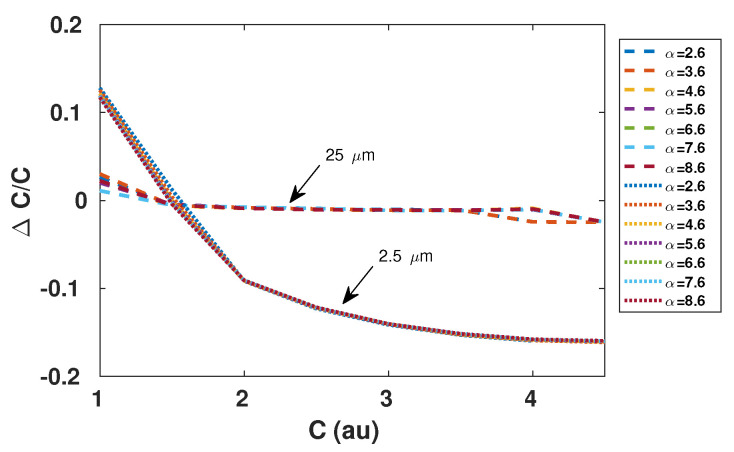
Simulated relative error in estimation of *C* based on the third harmonic amplitude of the reconstructed displacement using the non-uniform sampling approach as described in [22], with spline interpolation, for different α values. Remote sinusoidal displacement is of 2.5 μm (plain line) and 25 μm (dash line) amplitude at 90 Hz, $\lambda_{0}$ = 785 nm, and sampling frequency of the SM signal is 1 MS/s.

**Figure 6 sensors-21-03528-f006:**
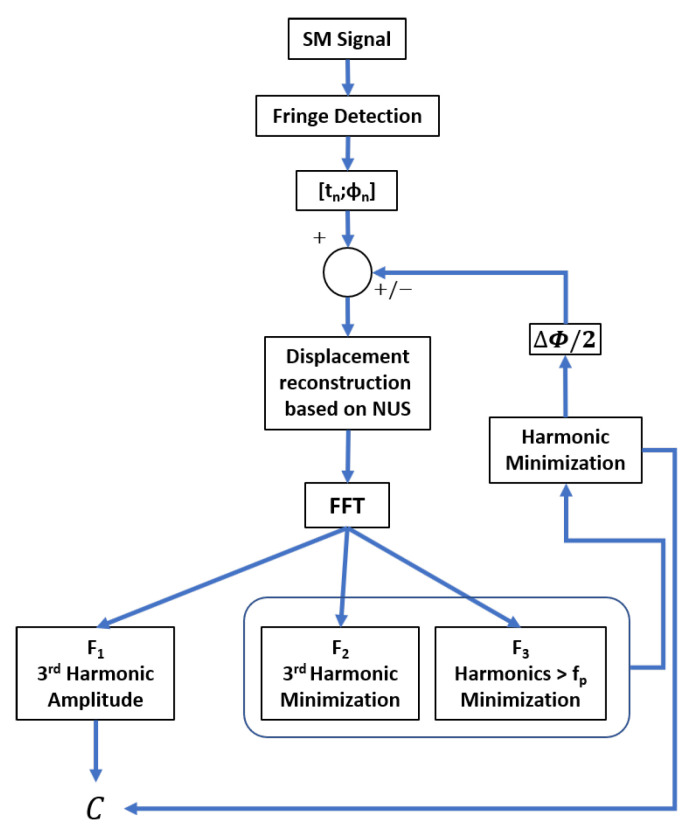
Block diagram of the proposed approaches to estimate *C*.

**Figure 7 sensors-21-03528-f007:**
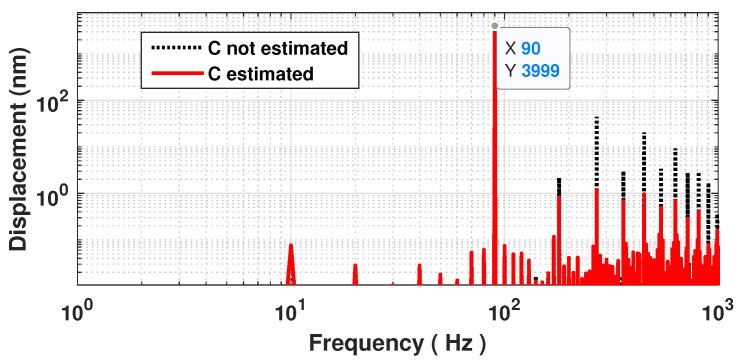
Simulated spectrum of a reconstructed displacement with 4 μm amplitude at 90 Hz by processing a SM signal with *C* = 3: without estimating *C* (dashed black line) and with estimating *C* and correcting the phase accordingly (plain red line).

**Figure 8 sensors-21-03528-f008:**
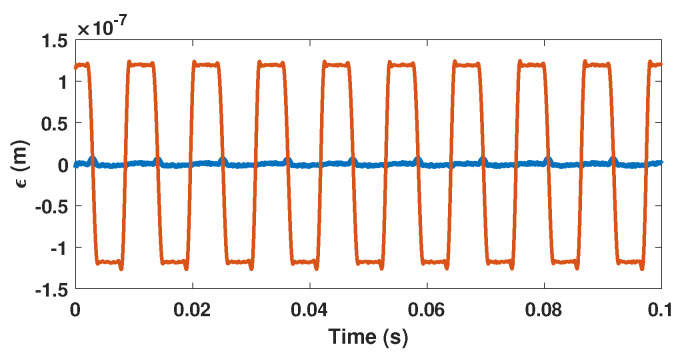
Simulated displacement reconstruction error ϵ for a displacement of 4 μm amplitude at 90 Hz with *C* = 3.3 without estimating *C* (orange line) and with estimating *C* and correcting the phase accordingly (blue line) using $F_{2}$.

**Figure 9 sensors-21-03528-f009:**
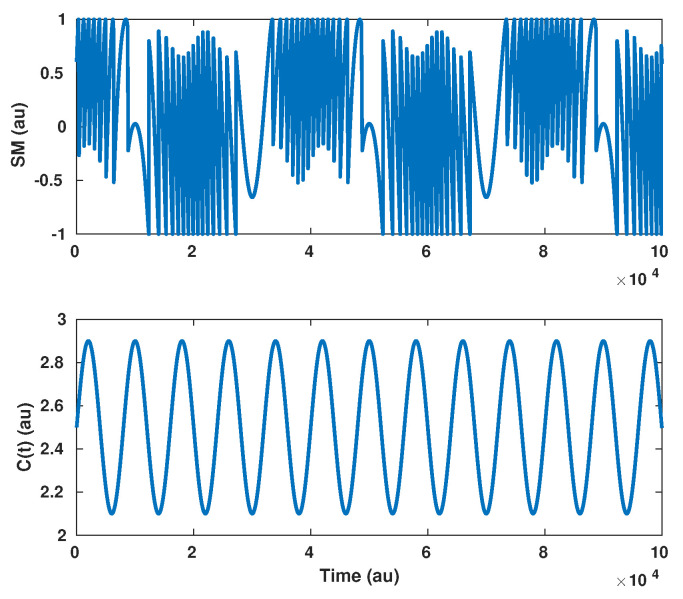
Simulated SM signal for a 3 μm amplitude at $f_{0}$ = 25 Hz displacement with $C=2.5+0.4\times \sin\left(2\pi \times 5f_{0}t\right)$.

**Figure 10 sensors-21-03528-f010:**
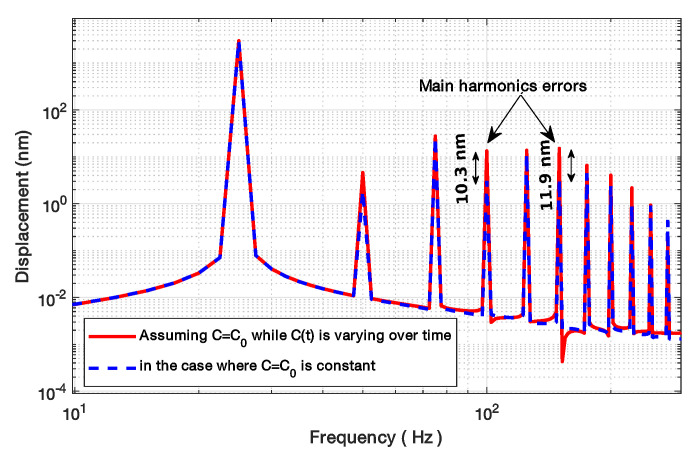
Simulated spectrum of the reconstructed displacement from the SM signal described in Figure 9, where *C* varies periodically, assuming that *C* is constant (red line), while the blue curve shows the spectrum of the reconstructed displacement if *C* is constant and equal to 2.5.

**Figure 11 sensors-21-03528-f011:**
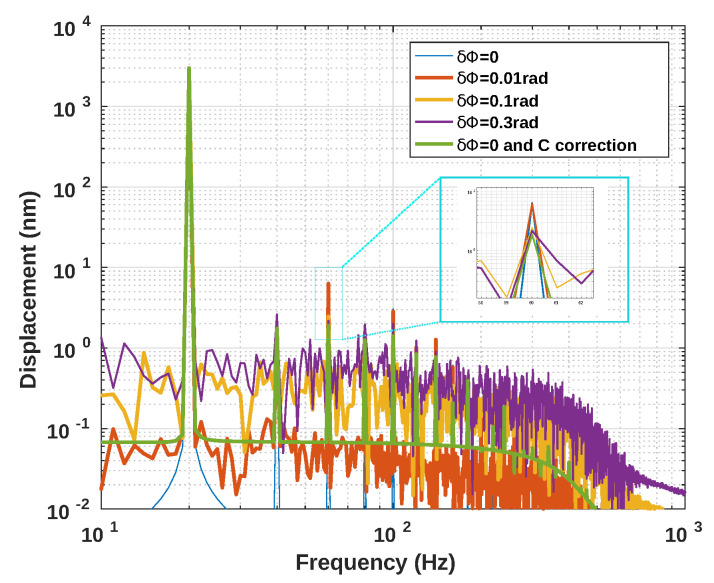
Simulated 3 μm amplitude sinusoidal displacement at 20 Hz reconstructed, using spline interpolation, from an SM signal with *C* = 1.5 and different phase noise levels δΦ: 0 (with and without *C* phase correction), 0.01 rad, 0.1 rad, 0.3 rad.

**Figure 12 sensors-21-03528-f012:**
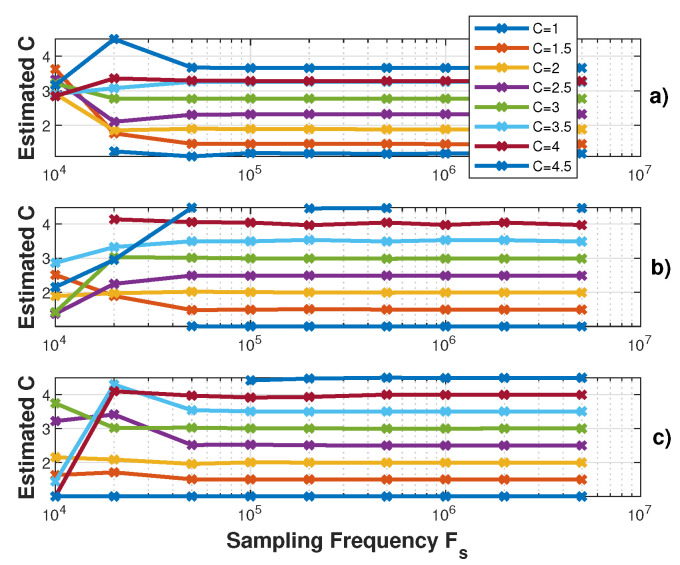
Simulated estimated *C* vs. the sampling frequency $f_{s}$ for a displacement of 3 μm amplitude at 20 Hz with *C* varying from 1 to 4.5 by 0.5 steps and α=3.6: (**a**) using the proposed algorithm $F_{1}$, (**b**) $F_{2}$ and (**c**) $F_{3}$.

**Figure 13 sensors-21-03528-f013:**
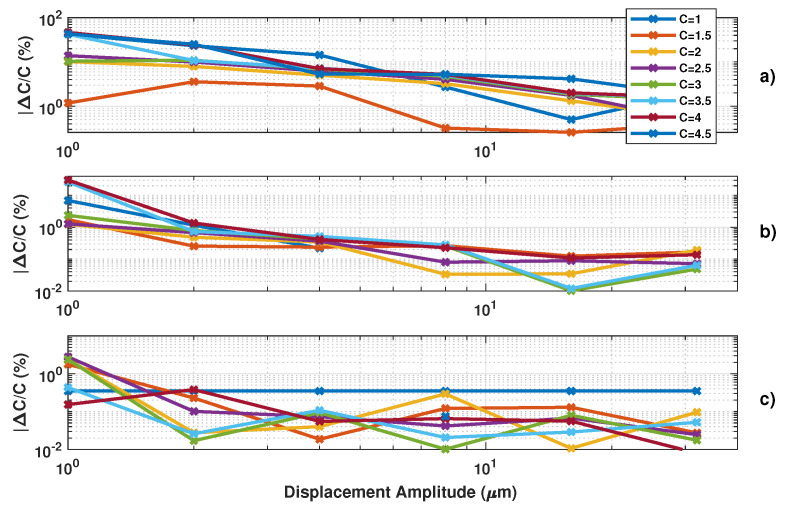
Simulated estimated *C* vs. displacement amplitude at 20 Hz with *C* varying from 1 to 4.5 by 0.5 steps and α=3.6 ($f_{s}\;=\;1\;$MHz): (**a**) using the proposed algorithm $F_{1}$, (**b**) $F_{2}$ and (**c**) $F_{3}$.

**Figure 14 sensors-21-03528-f014:**
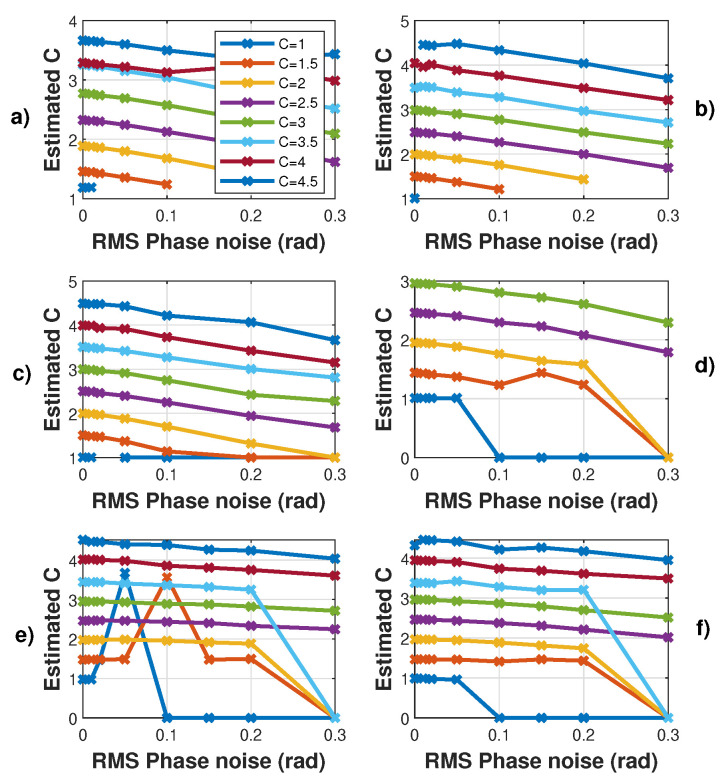
Simulated estimated *C* vs. RMS phase noise for a displacement of 3 μm amplitude at 20 Hz with *C* varying from 1 to 4.5 by 0.5 steps and α=3.6: (**a**) using the proposed algorithm $F_{1}$, (**b**) $F_{2}$ and (**c**) $F_{3}$, (**d**) using [19], (**e**) [21] and (**f**) using [8]. Note that each time a “0” value occurs, it corresponds to an absence of the algorithm convergence.

**Figure 15 sensors-21-03528-f015:**
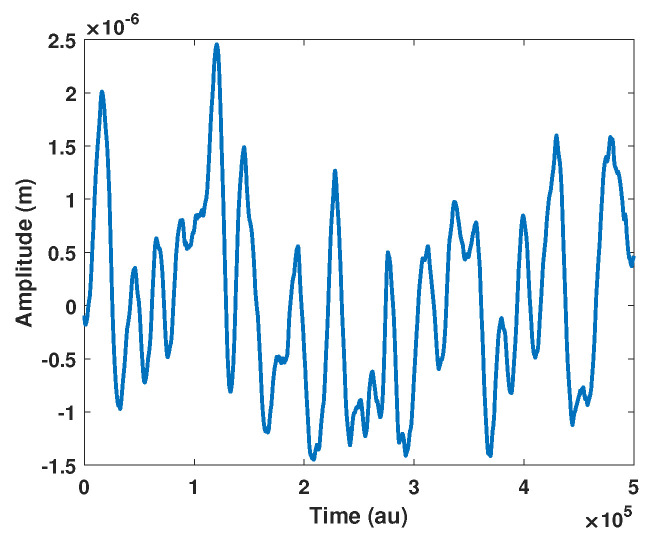
Simulated arbitrary displacement.

**Figure 16 sensors-21-03528-f016:**
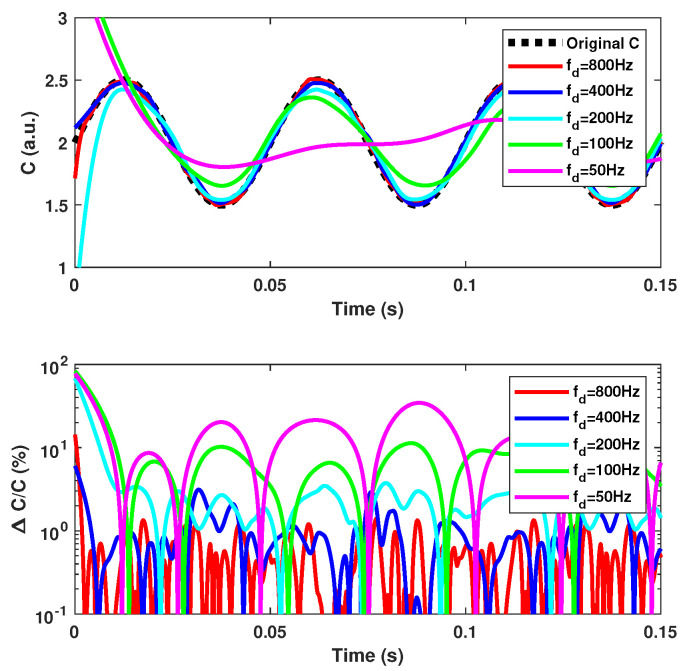
Simulated time-varying *C* estimation vs. the dithering frequency $f_{d}$ using method $F_{3}$ with the Hanning sliding window spectral analysis obtained for a target vibration at 10 Hz with a 5 μm amplitude, while *C* varies sinusoidally at $f_{C}$ = 20 Hz with an amplitude of 0.5 and offset of 2. The dithering signal corresponds to an equivalent vibration of 2 μm amplitude at $f_{d}$. The sliding window span is 5/$f_{d}$.

**Figure 17 sensors-21-03528-f017:**
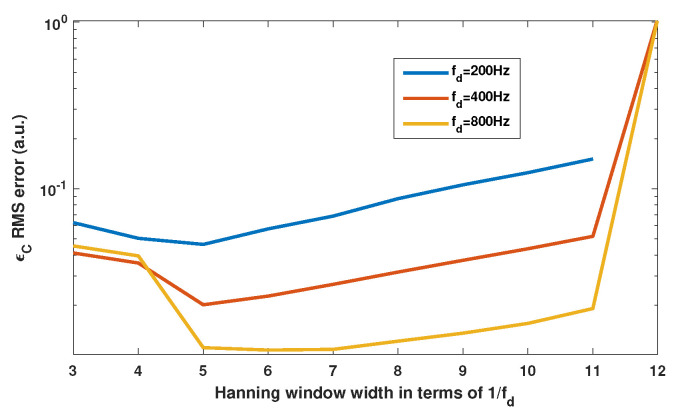
Simulated RMS error $\epsilon_{C}$ of the *C* estimation vs. the Hanning window size expressed as a multiple of dithering period $T_{d}$ = 1$/f_{d}$. Here, the method $F_{3}$ with the Hanning sliding window spectral analysis is applied on the SM signal obtained for a target vibration at 10 Hz with a 5 μm amplitude, while *C* is varying sinusoidally at $f_{C}$ = 20 Hz with an amplitude of 0.5 and offset of 2. The dithering signal corresponds to an equivalent vibration of 2 μm amplitude at $f_{d}$.

**Figure 18 sensors-21-03528-f018:**
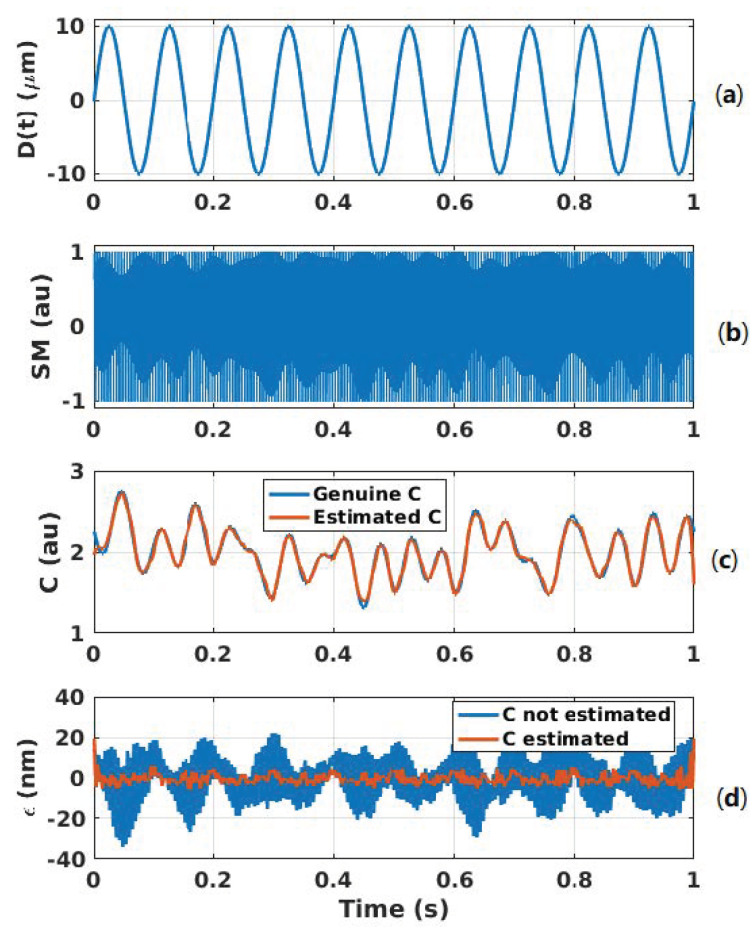
Simulated displacement reconstruction error ϵ for a displacement of 10 μm amplitude at 10 Hz with *C* varying: (**a**) displacement $D\left(t\right)$, (**b**) SM signal with a 2 μm dithering signal at 200 Hz, (**c**) estimated *C* (orange line) compared to genuine *C* (blue line) and (**d**) error ϵ in the case of *C* estimation and corresponding phase correction (orange line) and without *C* estimation (blue line).

**Figure 19 sensors-21-03528-f019:**
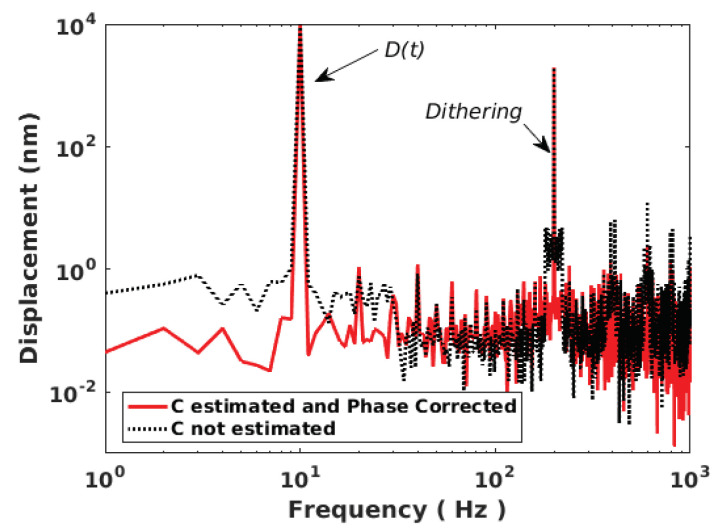
Simulated reconstructed displacement spectrum for a displacement of 10 μm amplitude at 10 Hz with *C* varying arbitrarily (limited to 20 Hz bandwith) obtained with 2 μm dithering signal at 200 Hz: with estimating *C* and correcting the phase accordingly (red line) and not correcting it (dashed black line).

**Figure 20 sensors-21-03528-f020:**
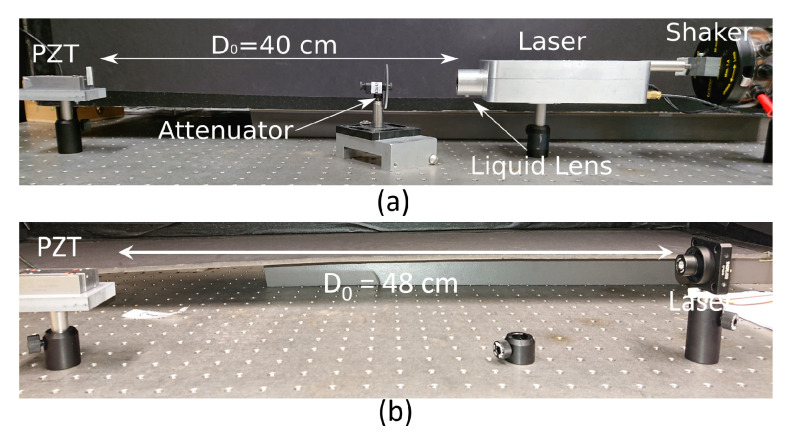
The *C* estimation test bench: (**a**) TB1 with a laser diode at $\lambda_{0}$ = 785 nm and (**b**) TB2 with a laser diode at $\lambda_{0}$ = 1550 nm.

**Figure 21 sensors-21-03528-f021:**
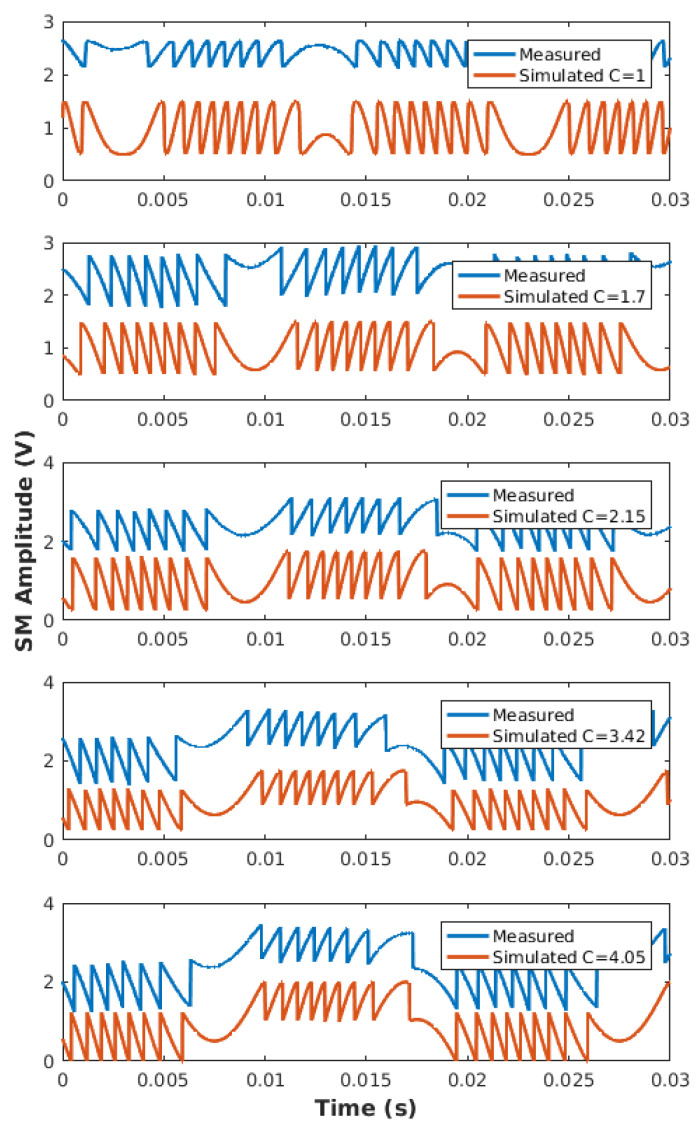
Comparison between the measured SM signals (Table 3) and simulated SM signal obtained with the estimated *C* value for a similar displacement amplitude.

**Figure 22 sensors-21-03528-f022:**
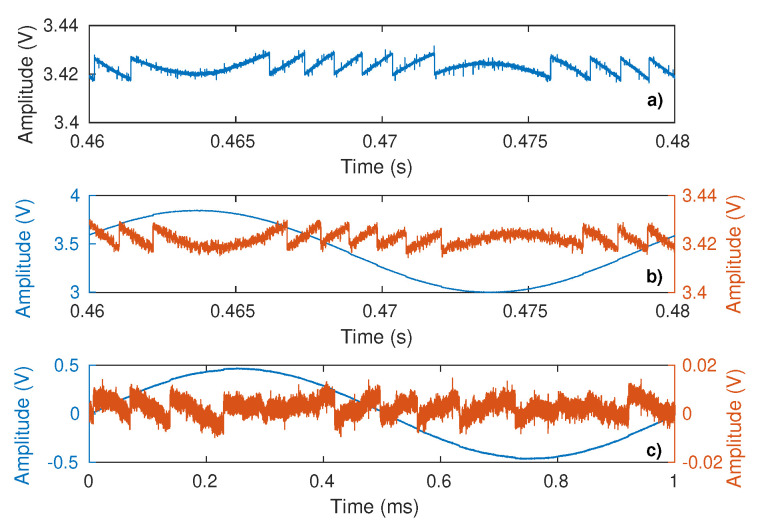
Measured SM signal: (**a**) without current modulation with target vibrating at 50 Hz, (**b**) with current modulation at 50 Hz (blue) and after power modulation removal (red), and (**c**) with current modulation at 1 kHz (blue) and after power modulation removal (red).

**Figure 23 sensors-21-03528-f023:**
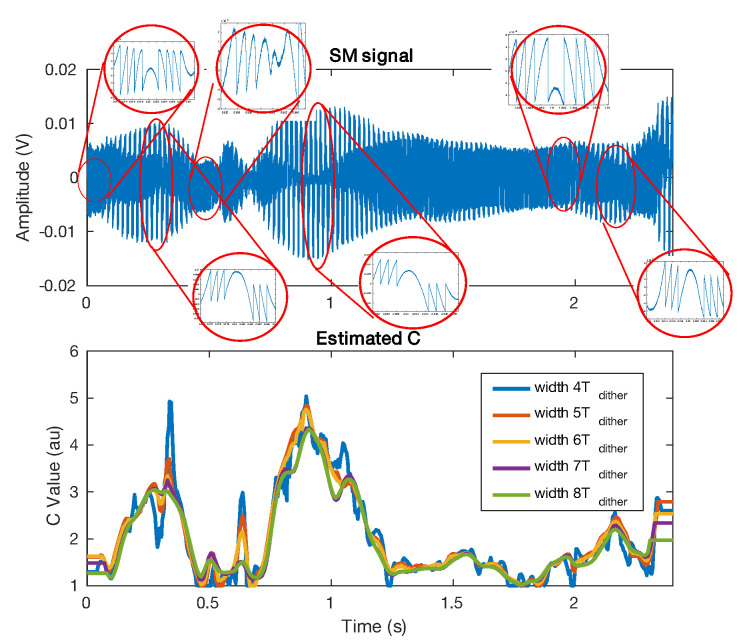
Measured *C* from a speckle affected SM signal for a target vibrating at 50 Hz. The *C* value was estimated using the method proposed in Sub-Section III.D for different Hanning window sizes.

**Figure 24 sensors-21-03528-f024:**
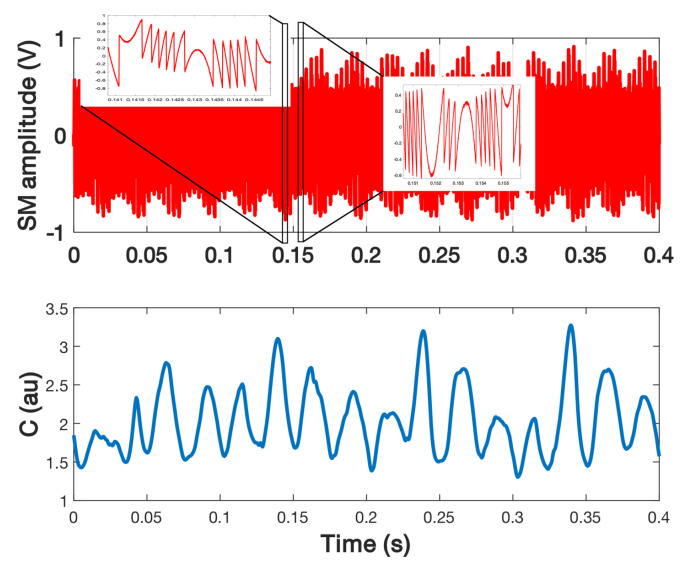
Measured *C* from a SM signal obtained for a loudspeaker vibrating at 40 Hz with a 2.4 μm amplitude and a dithering signal generated by a shaker at 290 Hz with a 1 μm amplitude.

**Figure 25 sensors-21-03528-f025:**
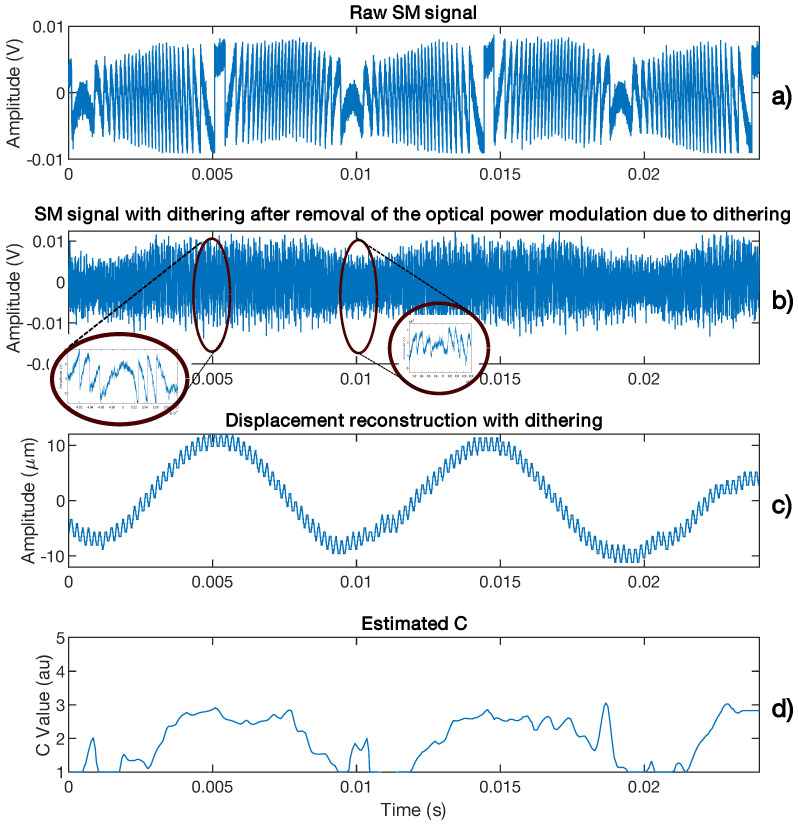
Measured *C* from a speckle affected SM signal for a target vibrating at 170 Hz: (**a**) without applying any dither in order to ease the visualization of the variation of *C*, (**b**) SM signal obtained for the same displacement and by applying current dithering at 5 kHz, (**c**) reconstructed displacement showing clearly the presence of dithering at 5 kHz and (**d**) the estimated *C* from (**b**). The *C* value is estimated by using the method proposed in Section 3.4.

**Table 1 sensors-21-03528-t001:** Estimated *C* with the proposed method and with the ones proposed in [8,19,21] for a simulated 3 μm sinusoidal displacement at 20 Hz sampled at $f_{s}$ = 1 MS/s with α = 3.6 without noise.

*C*	Our Work			[19]	[21]	[8]
	*F* _1_	*F* _2_	*F* _3_			
0.5	N/A	N/A	N/A	N/A	0.613	0.488
1	1.184	1.046	1.003	1.01	0.991	0.999
1.5	1.458	1.511	1.501	1.46	1.506	1.499
2	1.887	2.01	2.001	1.98	2.015	1.999
2.5	2.323	2.506	2.501	2.47	2.518	2.499
3	2.771	3.004	3.000	2.98	3.009	2.999
3.5	3.256	3.507	3.505	N/A	3.494	3.499
4	3.286	4.056	3.999	N/A	4.043	3.991
4.5	3.663	4.476	4.497	N/A	4.553	4.493
5	4.05	5.08	4.965	N/A	5.016	4.989
6	4.85	6.13	6.060	N/A	6.096	5.98
7	5.75	7.19	7.069	N/A	7.135	6.974

**Table 2 sensors-21-03528-t002:** Estimated *C* with the proposed method $F_{3}$ with and without dithering for a simulated arbitrary displacement (Figure 15) at $f_{s}$ = 1 MS/s with α = 3.6.

*C*	$F_{3}$	
	**No Dither**	**Dither**
1	1.422	1.003
1.5	1.529	1.491
2	1.743	1.998
2.5	1.605	2.503
3	1.976	3.005
3.5	2.649	3.521
4	1.997	4.013
4.5	2.582	4.526

**Table 3 sensors-21-03528-t003:** Measured *C* with the proposed method $F_{3}$ and with [8,19,21] using TB1. In total, 20 measurements were performed for each *C* value.

N	Our Work $F_{3}$		[19]		[21]		[8]	
	$\bar{C}$	$\sigma_{C}$	$\bar{C}$	$\sigma_{C}$	$\bar{C}$	$\sigma_{C}$	$\bar{C}$	$\sigma_{C}$
1	1.01	0.03	1.01 *	0 *	1.75	0.7	0.94	0.07
2	1.03	0.06	1.01 *	0 *	1.84	0.4	1.17	0.04
3	1.71	0.05	1.9	0.06	2.87	0.25	2.49	0.03
4	2.15	0.10	2.3	0.08	2.95	0.2	2.84	0.05
5	3.42	0.10	3.5	0.1	3.57	0.21	3.96	0.12
6	4.05	0.10	–	–	3.97	0.33	4.53	0.17

* For these experiments, the standard deviation is 0 and the estimated C is the same since the algorithm failed to estimate C when too close to 1. As a result, it always returns the same value.

**Table 4 sensors-21-03528-t004:** Measured *C* with the proposed method $F_{3}$ without current modulation (a) with a current modulation at 50 Hz (b) and 1  kHz (c) and with [19] (except for the 8th experiment marked with * obtained with [18]) using TB2.

N	(a)		(b)		(c)		[19]	
	$\bar{C}$	$\sigma_{C}$	$\bar{C}$	$\sigma_{C}$	$\bar{C}$	$\sigma_{C}$	$\bar{C}$	$\sigma_{C}$
1	1.21	0.13	1.23	0.3	1.32	0.45	1.18	0.09
2	1.57	0.02	1.54	0.09	1.49	0.24	1.48	0.04
3	1.87	0.09	1.78	0.06	2.21	0.17	1.86	0.05
4	2.08	0.04	2.21	0.07	2.38	0.13	2.07	0.05
5	2.30	0.07	2.32	0.03	2.51	0.18	2.342	0.04
6	2.79	0.07	2.75	0.05	3.01	0.31	2.70	0.07
7	3.31	0.10	3.30	0.12	3.48	0.39	3.23	0.08
8	4.65	0.14	4.19	0.05	4.27	0.35	4.24 *	0.02

## Data Availability

Not applicable.

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
