# Peer review of "Toward an Estimation of the Optical Feedback Factor C on the Fly for Displacement Sensing"

_sensors, 2021, doi:10.3390/s21103528_

Round 1

Reviewer 1 Report

The author reported a method to estimate the optical feedback factor C of laser diode. It is well organized with appropriate structure, experiment and analysis. The method will be interesting to the readers of optical community. However, some questions need to be clarified as following:

  1. There are already some methods to estimate the factor C in laser feedback field. Could the authors give a comparison between them?
  2. In my opinion, to correctly estimate the factor C, the feedback level should be strictly controlled. Otherwise, the error will rise sharply. How about the method proposed?
  3. How about using the EOM or AOM to replace the PZT?

Reviewer 2 Report

The manuscript contains novel and interesting results about data processing methods aiming at reconstructing the displacement of a target from "optical feedback interferometry" or equivalently "self-mixing interferometry". Contrary to many methods which relie on potentially complex hardware, the proposed approach consists in recovering the numerical value of the C parameter of the Lang-Kobayashi equations, widely used for modelling of this kind of experiments. Doing so, it reinforces one of the key strengths of the optical feedback interferometry approach, which is its hardware simplicity.

The paper is well written and presents new and interesting results obtained from an extensive simulation and experimental work. However, some points should be addressed by the authors before I can recommend publication of the work.

1) The introduction is well written and close to exhaustive in mentioning many approaches aiming at recovering values of alpha and C, parameters which are often used in physical modelling and phase unwrapping. To make it more complete, the authors should consider reference [https://doi.org/10.1364/OE.419844] where the estimation of these parameters is shown not to be the only possible approach, including sub-wavelength displacement.

2) The proposed method builds on the coexistence of multiple external cavity modes which are stable for a single paramter value, a multistability situation which is illustrated by the presence of hysteresis in response to a very simple (slow and periodic displacement). Would the method be robust to dynamical noise or to a very fast and sudden displacement causing switching of the laser from one external cavity mode to another, perhaps five or eight wavelengths away from the initial situation? These points should be discussed at least briefly.

3) At pp. 12-13 the authors describe how their approach can be applied to arbitrary displacement, leading to much better estimations of C than previous methods. This is an important strength of the method and it should be discussed some more: the parameters of the arbitrary displacement should be given (perhaps a spectral bandwidth for instance). In addition, the method should be applied to the end, _ie_ not only to measure C but to actually reconstruct the displacement of a target, especially experimentally.

4) The authors claim on the basis of numerical simulations that "the displacement reconstruction RMS error can be reduced by a factor of approximately 20, down to 2.5 nm". However, in the experiments reported on figure 25, the displacement is periodic but the first maximum and the third maximum of the reconstructed displacement differ by about 3 micrometers. That means that while the theoretical error is 2.5nm, the experimental one is three orders of magnitude larger. Perhaps the authors could add a trace of the known displacement to figure 25c) so that it can be compared to the reconstruction. Certainely, the authors should discuss the origin of this very large error (beyond "some artifacts that are most probably
induced by some missed fringe detections") and how this problem can be corrected so that the displacement measurement can fully benefit from the accurate on-the-fly determination of C.

5) In the conclusion, the merits of the proposed method are discussed in comparison to other methods of recovering C. This discussion is accurate, well written and interesting. However, since the point of the study is to recover C to measure displacements (down to 2.5nm precision stated in the introduction), the conclusion should be completed by a discussion of the actual performance in displacement reconstruction, perhaps in relation to alternative displacement reconstruction approaches. 

Round 2

Reviewer 2 Report

The authors have addressed my comments. Perhaps the abstract should be modified to "Most of the phase unwrapping open-loop OFI algorithms rely on the estimation of C to retrieve the displacement" instead of "All the phase unwrapping open-loop OFI algorithms rely on the estimation of C to retrieve the displacement".